# Infant Mode of Delivery Shapes the Skin Mycobiome of Prepubescent Children

Yan-Ren Wang,[a] Ting Zhu,[a] Fan-Qi Kong,[b] Yuan-Yuan Duan,[b] Carlos Galzote,[c] Zhe-Xue Quan[a]

[a]Ministry of Education Key Laboratory for Biodiversity Science and Ecological Engineering, Institute of Biodiversity Science, School of Life Sciences, Fudan University, Shanghai, China
[b]AP Skin Testing Center, Johnson & Johnson China Ltd., Shanghai, China
[c]Johnson & Johnson International (Singapore) Pte. Ltd., Singapore, Singapore

Yanren Wang and Ting Zhu contributed equally to this work. The order was determined by the corresponding author after negotiation.

**ABSTRACT** Characterizing the skin mycobiome is necessary to define its association with the host immune system, particularly in children. In this study, we describe the skin mycobiome on the face, ventral forearm, and calf of 72 prepubescent children (aged 1 to 10 years) and their mothers, based on internal transcribed spacer (ITS) amplicon sequencing. The age and delivery mode at birth are the most influential factors shaping the skin mycobiome. Compared with that of the vaginally born children, the skin mycobiome of caesarean-born children is assembled by predominantly deterministic niche-based processes and exhibits a more fragile microbial network at all three sampling sites. Moreover, vaginal delivery leads to clearer intra- and interindividual specialization of fungal structures with increasing age; this phenomenon is not observed in caesarean-born children. The maternal correlation with children also differs based on the mode of delivery; specifically, the mycobiomes of vaginally born children at younger ages are more strongly correlated with vagina-associated fungal genera (*Candida* and *Rhodotorula*), whereas those of caesarean-delivered children at elder age include more skin-associated and airborne fungal genera (*Malassezia* and *Alternaria*). Based on this ecological framework, our results suggest that the delivery mode is significantly associated with maturation of the skin fungal community in children.

**IMPORTANCE** Human skin is permanently colonized by microbes starting at birth. The hygiene hypothesis suggests that a lack of early-life immune imprinting weakens the body's resilience against atopic disorders later in life. To better understand fungal colonization following early-life periods affected by interruption, we studied the skin mycobiomes of 73 children and their mothers. Our results suggest a differentiation of the skin mycobiomes between caesarean-born and vaginally born children. Caesarean-born children exhibit a mycobiome structure with more fitted deterministic niche-based processes, a fragile network, and an unchanged microbial dissimilarity over time. In vaginally born children, this dissimilarity increases with age. The results indicate that initial microbial colonization has a long-term impact on a child's skin mycobiome. We believe that these findings will inspire further investigations of the "hygiene hypothesis" in the human microbiome, especially in providing novel insights into influences on the development of the early-life microbiome.

**KEYWORDS** skin mycobiome, fungi, prepubescent children, delivery mode

Human skin serves as the first line of defense against pathogens and simultaneously acts as a nursery for diverse microbes, comprising bacteria, fungi, archaea, and viruses (1 to 3). Moreover, cutaneous fungi have emerged as potent inducers of antigen-specific T cells in humans, engaging in homeostatic host–fungus interactions and

Address correspondence to Zhe-Xue Quan, quanzx@fudan.edu.cn.

The authors declare a conflict of interest. Y.-Y.D., F.-Q.K., and C.G. were employed by Johnson & Johnson. The remaining authors declare that the research was conducted in the absence of any commercial or financial relationships that could be construed as potential conflicts of interest. The authors declare that this study received funding from Johnson & Johnson International Pte Ltd. (Singapore). The funder was not involved in the study design, collection, analysis, interpretation of data, writing of this article, or the decision to submit it for publication.

contributing to immune pathology when dysregulated (4). Commensal and pathogenic fungi in the skin are associated with human health, infections, and diseases, including acne vulgaris (5, 6), tinea pedis (7), and dandruff (8, 9). Thus, understanding the determinants of skin microbial stability is pivotal in defining the contribution made by homeostatic functions in microbiota, and their corresponding impact on host health (3).

The skin physiology of infants (10) and children (11) is distinct from that of adults, with a thinner stratum corneum, lower production of natural moisturizing factors and skin lipids, and a more immature host immune system (12). These factors increase the risk of prepubescent children developing acute irritation and infections by opportunistic fungal pathogens (13, 14). Moreover, birth transitions humans from a relatively microbe-free (15) aqueous uterus to a microbe-rich atmospheric environment. Neonatal skin is colonized by microbes from maternal and nonmaternal sources (e.g., a hospital environment), which contribute to local and systemic immune development (16). However, microbial exposure during this period alters the skin immune trajectory as well as disease susceptibility (17). For example, delayed bacterial development caused by exoteric interruptions (e.g., use of antibiotics, caesarean delivery, or diet) in the gut microbiome of infants (18) can have irreversible consequences, along with a higher incidence of several diseases (e.g., obesity, asthma, allergies, and infection) in early childhood (19 to 22), adolescence (23, 24), and adulthood (25, 26). A compelling explanation for these phenomena is a dysbiosis between commensal microbes and the host immune system (27).

The delivery method significantly impacts the microbial colonization of infant skin (28 to 31) and influences resident bacteria in prepubescent children (32). However, studies examining the skin mycobiome in children, especially regarding maternal influence, have been limited primarily to children aged 6 months or less (28, 33). Accordingly, we previously investigated the effect of the delivery mode on the skin bacterial population of prepubescent children and found that it lasted for 10 years (32). Accordingly, in the current study, we have expanded the analysis scope to include the skin mycobiome of children aged 1 to 10 years and analyzed the contributing factors and the maturational dynamics of cutaneous fungi in children.

## RESULTS

**Summary of analyzed samples and sequencing data.** We included 432 skin mycobiome samples from the face, calf, and ventral forearm of 72 children, aged 1 to 10 years, and their mothers. Demographic information is presented in Table S1 in the supplemental material. In total, 3,230,944 and 2,922,054 high-quality ITS1 sequences were obtained from the samples of children and mothers, respectively, with an average of 14,208 sequence reads per sample and a minimum of 2,565 sequence reads. All sequences were clustered into operational taxonomic units (OTUs) according to a 95% similarity cutoff. OTUs that only appeared in a single sample or had an extremely low relative abundance (<0.01%) were removed, resulting in a total of 512 OTUs for further analysis.

**Skin mycobiome profiles from three sampling sites on prepubescent children.** The PERMANOVA results (Table 1) identified age (pseudo-$F$ = 3.000, $P \leq 0.05$) as the most significant driving factor of the skin mycobiome, followed by delivery mode (pseudo-$F$ = 2.136, $P \leq 0.05$, including children delivered through vaginal birth and caesarean section), and birthplace (pseudo-$F$ = 2.051, $P \leq 0.05$, including children born in suburban and urban areas). However, sex did not have a significant effect. Next, 216 samples from children were divided into 36 subgroups according to the sampling site, participant age, and delivery mode. Principal coordinate analysis (Fig. S1) of the fungal community was then performed for these 36 subgroups. The group of 10-year-old children formed a distinct cluster (Fig. S1a); the groups based on different delivery modes differed by axis 1, and vaginally born children were more alike than their caesarean-born counterparts (Fig. S1b), whereas groups for the different sampling sites did not exhibit specific clustering (Fig. S1c).

The main fungal genera detected at the three sampling sites from the children (Fig. 1a) were *Malassezia* (15.11%), *Cladosporium* (10.29%), *Alternaria* (6.62%), *Candida* (6.04%), and

**TABLE 1** Permutational multivariate analysis of variance assessing Bray-Curtis dissimilarity between children's groups in different demographics[a]

| Factor | Sample property[b] | pseudo-$F$ value of PERMANOVA | | | | | |
|---|---|---|---|---|---|---|---|
| | | Age | Delivery mode | Birthplace | Site | Feeding | Sex |
| Age, pseudo-$F$ = 3.000 | Age 1, $n$ = 39 | | 2.662 | 1.837 | 1.136[NS] | 1.405 | 1.379[NS] |
| | Age 2, $n$ = 30 | | 1.327[NS] | 1.205[NS] | 0.677[NS] | 1.704 | 1.190[NS] |
| | Age 3, $n$ = 36 | | 2.209 | 0.898[NS] | 1.051[NS] | 1.410[NS] | 1.112[NS] |
| | Age 4, $n$ = 42 | | 1.352[NS] | 1.267[NS] | 0.964[NS] | 1.079[NS] | 1.323[NS] |
| | Age 5, $n$ = 39 | | 0.935[NS] | 1.354[NS] | 1.132[NS] | 1.142[NS] | 0.994[NS] |
| | Age 10, $n$ = 30 | | 1.429 | 0.787[NS] | 0.961[NS] | 1.316 | 1.385[NS] |
| Delivery mode, pseudo-$F$ = 2.136 | Vaginal delivery, $n$ = 132 | 2.534 | | 1.475 | 1.482 | 1.447 | 1.209[NS] |
| | C-section, $n$ = 84 | 1.927 | | 1.692 | 0.906[NS] | 1.140[NS] | 1.039[NS] |
| Birthplace, pseudo-$F$ = 2.051 | Suburb, $n$ = 129 | 2.126 | 2.194 | | 1.392 | 1.035[NS] | 2.238 |
| | Urban, $n$ = 87 | 1.910 | 1.504 | | 1.025[NS] | 1.401 | 0.811[NS] |
| Individual, pseudo-$F$ = 1.756 | | | | | | | |
| Sampling site, pseudo-$F$ = 1.493 | Face, $n$ = 72 | 1.59 | 1.340[NS] | 1.676 | | 0.909[NS] | 1.094[NS] |
| | Calf, $n$ = 72 | 1.557 | 1.370[NS] | 1.217[NS] | | 1.056[NS] | 0.718[NS] |
| | Ventral Forearm, $n$ = 72 | 1.603 | 1.757 | 1.097[NS] | | 0.819[NS] | 0.990[NS] |
| Feeding type, pseudo-$F$ = 1.362 | Formula fed, $n$ = 48 | 1.518 | 1.432 | 0.901[NS] | 0.826[NS] | | 1.676 |
| | Mixed fed, $n$ = 84 | 1.834 | 1.867 | 1.725 | 1.139[NS] | | 0.930[NS] |
| | Breast fed, $n$ = 84 | 2.326 | 1.867 | 1.615 | 0.978[NS] | | 1.605 |
| Sex, $F$ = 1.398 [NS] | Female, $n$ = 90 | 1.959 | 1.982 | 1.216[NS] | 1.012[NS] | 1.217[NS] | |
| | Male, $n$ = 156 | 2.255 | 1.547 | 2.576 | 1.168[NS] | 1.642 | |

[a]Pseudo-$F$ values are considered a measure of the effect size of the difference. $P \leq 0.05$ for all presented pseudo-$F$ values except for numbers with superscript NS (no significance).
[b]$n$ indicates number of participant children.

*Rhodotorula* (5.57%). Moreover, the Shannon indices for the calf were significantly lower than those for the face and ventral forearm (both $P \leq 0.0001$, Wilcoxon test; Fig. 1b); however, no such difference was detected in the Chao1 indices (Fig. 1c). Additionally, the individual specificity of the calf was significantly higher than that of the other sites ($P \leq 0.0001$, Wilcoxon test; Fig. 1d). Furthermore, we examined the relationship between the skin's physical parameters and fungal community structure (Table S2) or the main

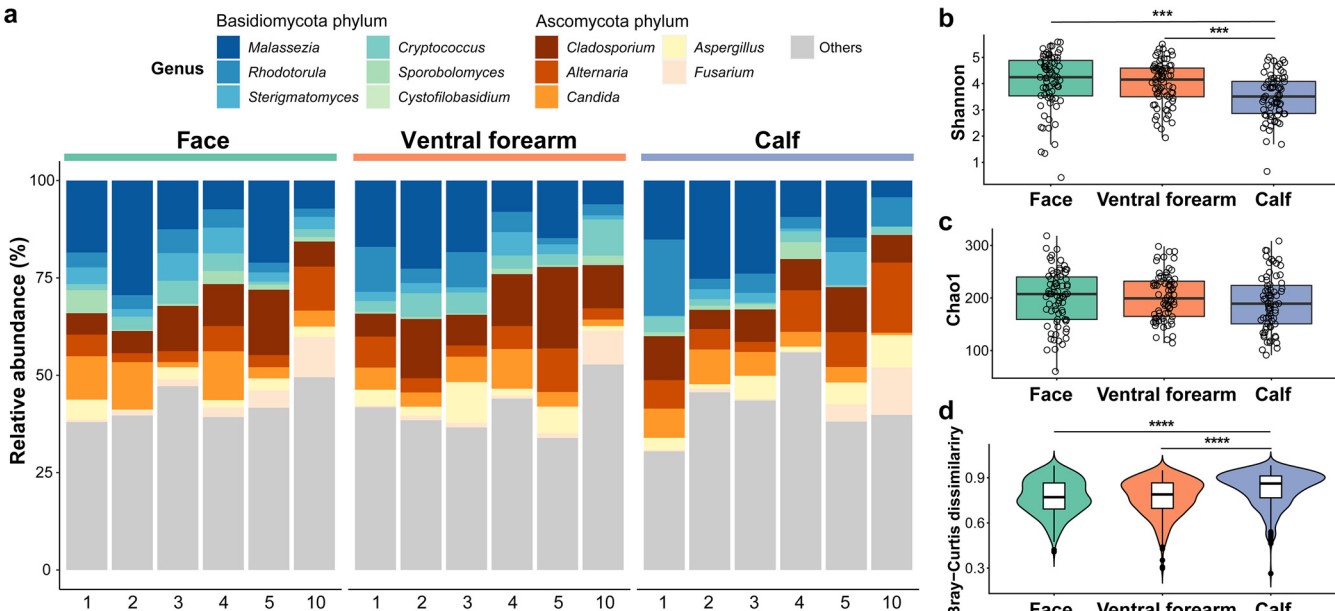

**FIG 1** Overall profile of the skin mycobiome in children ($n$ = 72). (a) Relative abundance of the main fungal genera (mean relative abundance > 1%) in children's samples collapsed by age and skin site. Numbers (1, 2, 3, 4, 5, and 10) represent children's age. Same-color schemes indicate the phylum level of the genera. (b to d) Fungal alpha diversity index (b: Shannon; c: Chao1) and beta diversity (d: Bray-Curtis dissimilarity) on the face, ventral forearm, and calf of children, as examined by Wilcoxon test. $P$ values of the Wilcoxon test are indicated as asterisks. NS presents no significance ($P$ > 0.05); ***, $P \leq 10^{-3}$; ****, $P \leq 10^{-7}$.

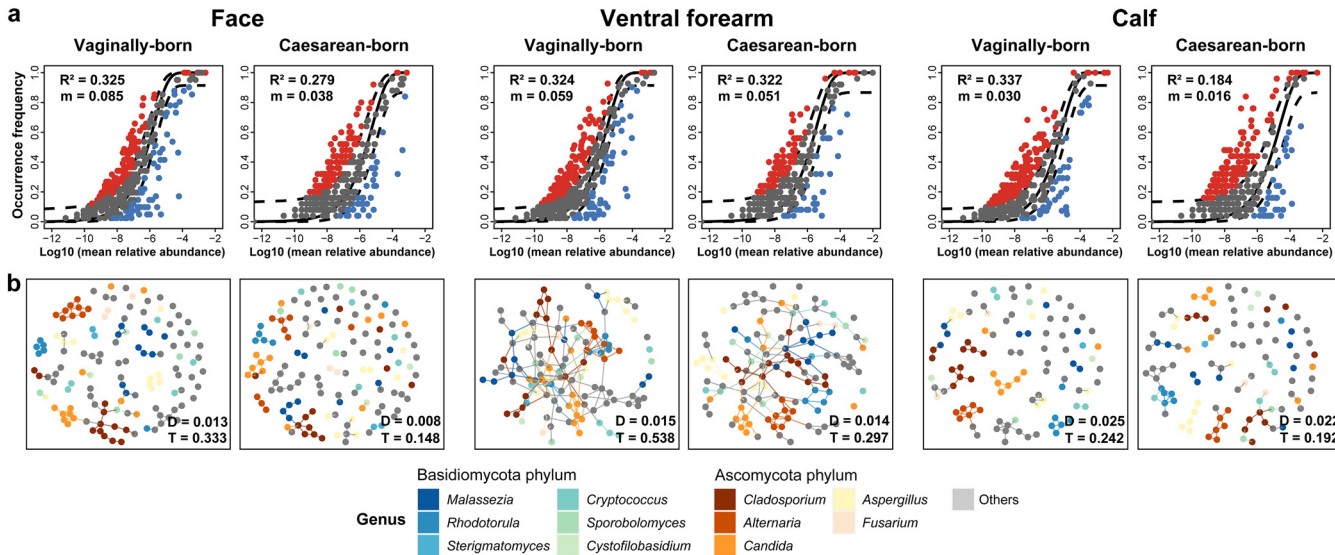

**FIG 2** Sloan neutral model predictions and network analysis of skin mycobiome. (a) Sloan neutral model prediction of the skin (including the face, ventral forearm, and calf) mycobiome from vaginally born ($n = 44$) and caesarean-born ($n = 28$) children. Dots represent the OTUs, and color represents whether the OTUs fitted above (red), within (gray), or below (blue) the 95% confidence interval (dotted lines) of the neutral model prediction. $R^2$ values (fitness to neutral assembly process) and m values (estimated migration rate) are shown. (b) Network analysis of skin fungal OTUs from vaginally born and caesarean-born children. Points represent the OTUs, and color represents the taxonomy affiliations at the genus level. D values (network density) and T values (network transitivity) are shown.

fungal genus composition (genera with mean relative abundance > 1%; Table S3). However, few significant correlations were observed (Tables S2 and S3).

**Neutral assembly process on skin mycobiomes and bacteriomes.** We applied the Sloan neutral prediction model to evaluate whether the microbial assembly process follows a stochastic (neutral model) or niche-based process (deterministic model) by fitting the occurrence frequency and mean relative abundance of microbial OTUs. The bacterial communities on skin were more influenced by dispersal (higher m values; fungi: $m = 0.048$; bacteria: $m = 0.491$; Fig. S2a) and showed a higher proportion of OTUs above the neutral prediction compared to fungi, although they were less fitted to the neutral model (fungi: $R^2 = 0.501$; bacteria: $R^2 = 0.489$; Fig. S2a). Core skin colonizers, including *Cladosporium* and *Propionibacterium*, were more prevalent in samples compared to those predicted by the neutral model (Fig. S2b). In contrast, *Sterigmatomyces* and *Acinetobacter* had a large proportion of OTUs below the neutral model (Fig. S2b), indicating that they were abundant in fewer samples than predicted.

**Influence of delivery mode on the skin mycobiomes of prepubescent children.** The skin mycobiomes of caesarean-born children did not differ significantly between sampling sites (vaginally born: pseudo-$F = 1.482$, $P \leq 0.05$; caesarean-born: pseudo-$F = 0.906$, $P > 0.05$) or feeding type (vaginally born: pseudo-$F = 1.447$, $P \leq 0.05$; caesarean-born: pseudo-$F = 1.140$, $P > 0.05$), whereas significant differences were observed in vaginally born children (Table 1). Moreover, the mycobiome richness (Chao1) was higher on the facial skin of vaginally born children ($P \leq 0.05$; Fig. S3a).

We further applied predictions of the fungal community assembly process using the Sloan neutral model and network analysis to assess ecological differences. The neutral model prediction results (Fig. 2a) showed that the data for vaginally born children exhibited a closer fit to the neutral model and were more influenced by dispersal than those for caesarean-born children in all groups, including samples from the face (vaginally born: $R^2 = 0.325$, migration rate [m] = 0.085; caesarean-born: $R^2 = 0.279$, $m = 0.038$), ventral forearm (vaginally born: $R^2 = 0.324$, $m = 0.059$; caesarean-born: $R^2 = 0.322$, $m = 0.051$), and calf (vaginally born: $R^2 = 0.337$, $m = 0.030$; caesarean-born: $R^2 = 0.184$, $m = 0.016$). Additionally, microbial network analysis (Fig. 2b) showed that the network density (D) and network transitivity (T, the probability that a node is interconnected to adjacent nodes) were higher in vaginally born children compared to caesarean-born

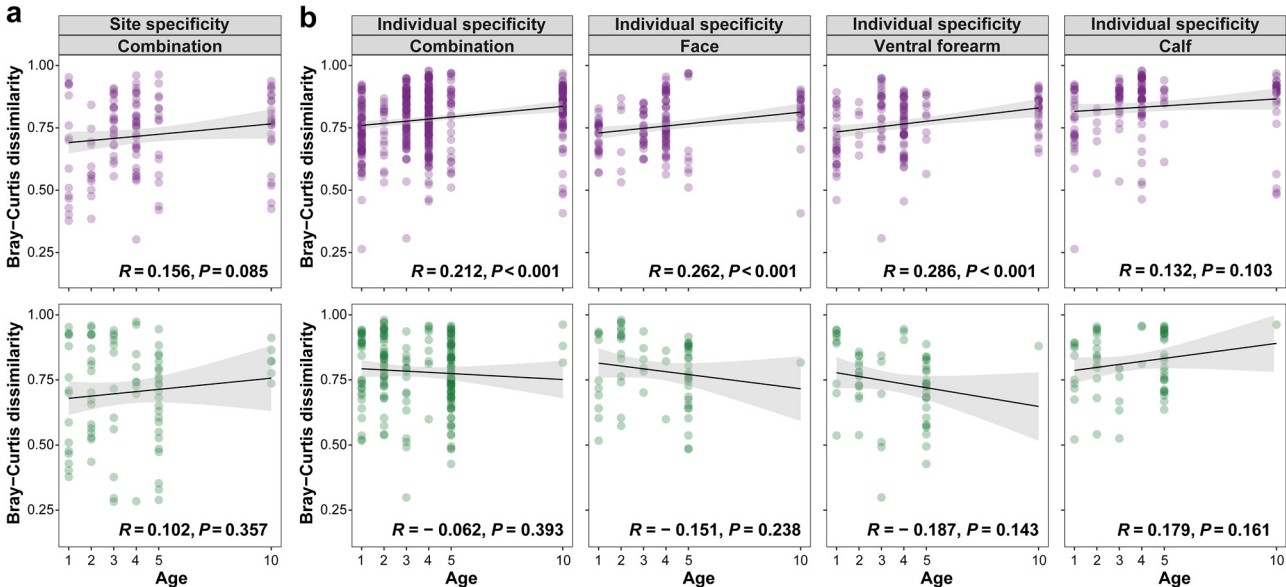

**FIG 3** Bray-Curtis dissimilarity trajectory of the skin mycobiome across children's age. (a) Site specificity, evaluated by Bray-Curtis dissimilarity of the mycobiome within different skin sites (intraindividual dissimilarity), from vaginally born and caesarean-born children. (b) Individual specificity, evaluated by Bray-Curtis dissimilarity of the mycobiome within different individuals (interindividual dissimilarity) of different skin sites in vaginally born and caesarean-born children. Points represent Bray-Curtis dissimilarity indices, containing skin samples of vaginally born (purple) and caesarean-born (green) children. R and P values of the Pearson's correlation test are shown.

children. This phenomenon was observed at all the sampling sites: face (vaginally born: D = 0.013, T = 0.333; caesarean-born: D = 0.008, T = 0.148), ventral forearm (vaginally born: D = 0.015, T = 0.538; caesarean-born: D = 0.014, T = 0.297), and calf (vaginally born: D = 0.025, T = 0.242; caesarean-born: D = 0.022, T = 0.192).

**Influence of delivery mode on maturation trajectory of the skin mycobiome of prepubescent children.** To determine whether the differences between the two delivery modes impact skin mycobiome maturation in children, we examined a linear regression of site specificity (intraindividual dissimilarity, i.e., the Bray-Curtis dissimilarity among different sites of one individual) and individual specificity (interindividual dissimilarity, i.e., the Bray-Curtis dissimilarity among different individuals of one sampling site) with increasing age. Generally, site specificity (R = 0.139, P = 0.042; Fig. S4a) and individual specificity (except for the face site, all $R \geq 0.123$, all $P \leq 0.010$; Fig. S4b) were positively correlated with the children's age. Furthermore, site specificity (Fig. 3a) in vaginally born children was more strongly correlated with age (vaginally born: R = 0.156, P = 0.085; caesarean-born: R = 0.102, P = 0.357). Likewise, individual specificity (Fig. 3b) from vaginally born children exhibited a significant positive correlation with age (vaginally born: R = 0.212, P < 0.001; caesarean-born: R = −0.062, P = 0.393); this correlation also was strong for specific sampling sites, namely, face (vaginally born: R = 0.262, P < 0.001; caesarean-born: R = −0.151, P = 0.238) and ventral forearm (vaginally born: R = 0.286, P < 0.001; caesarean-born: R = −0.187, P = 0.143), but not calf site (vaginally born: R = 0.132, P = 0.103; caesarean-born: R = 0.179, P = 0.161). Specifically, community dissimilarity within/between vaginally born children tended to be more diverse with increasing age, whereas a much weaker trend was observed in caesarean-born children.

Compared with caesarean-born children, 1-year-old vaginally delivered children exhibited greater relative abundances of several main fungal genera (Fig. S5): *Rhodotorula* (vaginally born: 15.73%; caesarean-born: 5.11%; $P_{fdr}$ = 0.2829), *Candida* (vaginally born: 10.97%; caesarean-born: 3.55%; $P_{fdr}$ = 0.1700), and *Cladosporium* (vaginally born: 10.52%; caesarean-born: 2.77%; $P_{fdr}$ = 0.0392). However, certain genera showed the opposite trend, namely, *Malassezia* (vaginally born: 13.31%; caesarean-born: 22.61%; $P_{fdr}$ = 0.4319), *Alternaria* (vaginally born: 4.78%; caesarean-born: 10.43%; $P_{fdr}$ = 0.4631), and *Aspergillus*

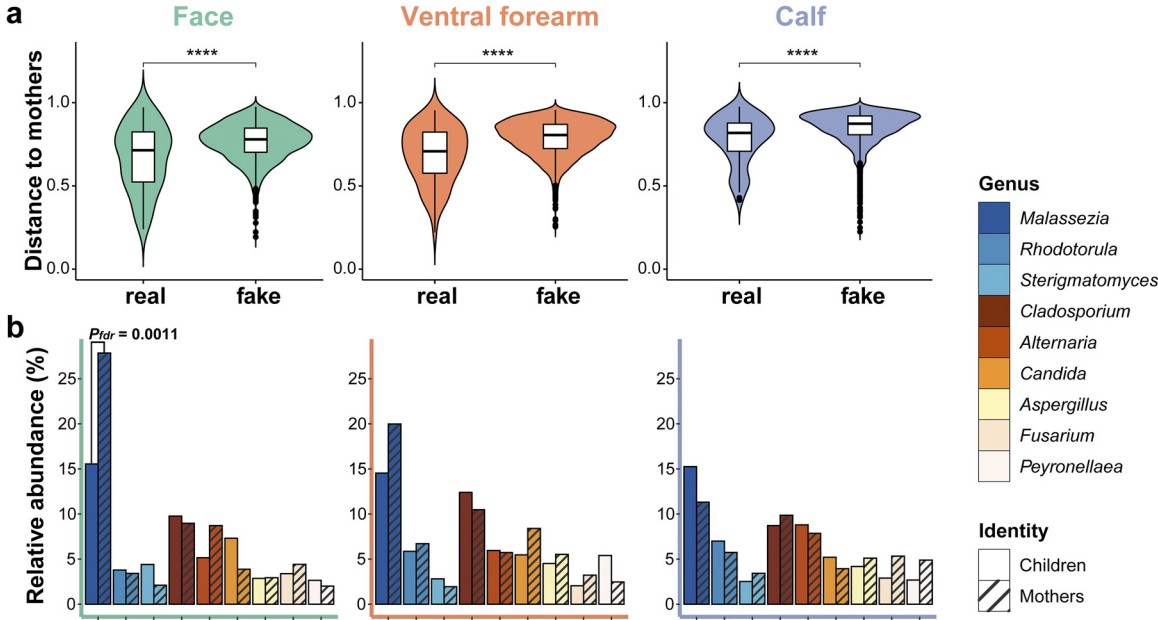

**FIG 4** Comparison of the skin mycobiome between children and mothers. (a) Violin plot of the Bray-Curtis dissimilarity of true mother–child pairs and false mother–child pairs in three different skin sites. (b) Main genera (mean relative abundance >3% on all samples) from three skin sites of mothers and children. Multiple Wilcoxon tests of all fungal genera in children and mothers are performed to adjust the $P$ value using the false discovery rate method. $P$ values of the Wilcoxon test in (a) are shown as asterisks (****, $P \leq 10^{-3}$); $P_{fdr}$ values of the Wilcoxon test in (b) are shown in numerical form.

(vaginally born: 2.86%; caesarean-born: 6.07%; $P_{fdr}$ = 0.8824). Furthermore, as the children's ages increased, *Candida* (vaginally born: R = −0.2248, $P_{fdr}$ = 0.1432; caesarean-born: R = −0.0971, $P_{fdr}$ = 0.9206) showed a more substantial trend of decrement in vaginally born children than in caesarean-born children (Table S4).

**Relationship between the skin mycobiomes of mothers and children.** The mothers of the 72 children were included for comparative analysis. Compared with the children, mothers' site specificity was stronger (mothers: pseudo-$F$ = 3.152, $P$ = 0.001; children: pseudo-$F$ = 1.493, $P$ = 0.001) than individual specificity (mothers: pseudo-$F$ = 1.616, $P$ = 0.001; children: pseudo-$F$ = 1.756, $P$ = 0.001). Mothers and children exhibited significant differences in skin fungal communities (pseudo-$F$ = 2.580, $P$ = 0.001); however, the alpha diversity of these two groups did not differ significantly. The Bray-Curtis dissimilarities of true mother–child pairs were significantly lower than those of false mother–child pairs (false: distance matrix data set of incorrect corresponding to mother–child pairs) at all three sampling sites (all $P \leq 0.05$; Fig. 4a).

Significant differences were observed in the relative skin mycobiome compositions of mothers and children for genus *Malassezia* at the face site (mothers: 27.85%; children: 15.56%; $P_{fdr}$ = 0.0011; Fig. 4b). Spearman's correlation test was then employed to reveal associations between the fungal genera abundance in mother–child pairs from two groups divided by children's age (Table 2 and Fig. S6; main genera with a mean relative abundance of >1% in all samples are presented). In groups with 1- to 3-year-old children, *Candida* (rho = 0.4067, $P_{fdr}$ = 0.0088) and *Rhodotorula* (rho = 0.3771, $P_{fdr}$ = 0.0167) showed a significant correlation between vaginally born children and their mothers but not between caesarean-born children and their mothers (rho $\leq$ 0.0505, $P_{fdr}$ $\geq$ 0.9161). Furthermore, vaginal-birth mother–child pairs from 4-, 5-, and 10-year-old children did not show significant correlations with respect to these two genera (rho $\leq$ 0.1253, $P_{fdr}$ $\geq$ 0.4684). Caesarean-birth mother–child pairs from the elder age group exhibited positive relationships with respect to *Malassezia* (rho = 0.5158, $P_{fdr}$ = 0.0104) and *Alternaria* (rho = 0.4555, $P_{fdr}$ = 0.0332); likewise, these trends were not present in the vaginal birth pairs from the elder group (rho $\leq$ 0.1306, $P_{fdr}$ $\geq$ 0.4640) and the caesarean birth pairs from the younger group (rho $\leq$ 0.2172, $P_{fdr}$ $\geq$ 0.5109). Additionally, in mother–child pairs of the

**TABLE 2** Spearman's correlation analysis between the mothers and children for different age groups and modes of delivery[a]

| Genus[b] | Group aged from 1 to 3 yrs old (n = 105) | | | | Group aged from 4 to 5, 10 yrs old (n = 111) | | | |
| | Vaginally born (n = 60) | | Caesarean-born (n = 45) | | Vaginally born (n = 72) | | Caesarean-born (n = 39) | |
| | Rho[c] | $P_{fdr}$ | Rho[c] | $P_{fdr}$ | Rho[c] | $P_{fdr}$ | Rho[c] | $P_{fdr}$ |
|---|---|---|---|---|---|---|---|---|
| Malassezia | 0.2485 | 0.1667 | 0.2172 | 0.5109 | 0.1306 | 0.4640 | 0.5158 | 0.0104 |
| Cladosporium | 0.1140 | 0.6070 | 0.3837 | 0.0751 | 0.2515 | 0.1121 | 0.3906 | 0.0767 |
| Alternaria | 0.1398 | 0.5207 | 0.0694 | 0.8923 | 0.0225 | 0.9144 | 0.4555 | 0.0332 |
| Candida | 0.4067 | 0.0088 | −0.0004 | 0.9978 | 0.1253 | 0.4854 | 0.2822 | 0.2746 |
| Rhodotorula | 0.3771 | 0.0167 | 0.0505 | 0.9161 | 0.1290 | 0.4684 | 0.1089 | 0.9062 |
| Aspergillus | 0.2298 | 0.0144 | 0.4846 | 0.0127 | 0.0860 | 0.6731 | 0.2022 | 0.5155 |
| Fusarium | 0.1700 | 0.3850 | −0.0634 | 0.8923 | 0.5731 | 6.292E−06 | 0.4270 | 0.0507 |
| Peyronellaea | 0.1920 | 0.3021 | 0.1074 | 0.8499 | 0.1171 | 0.5143 | 0.1273 | 0.8450 |
| Sterigmatomyces | 0.1300 | 0.5542 | 0.3779 | 0.0787 | 0.0491 | 0.8419 | −0.1602 | 0.6765 |
| Aureobasidium | 0.2905 | 0.0951 | 0.0376 | 0.9161 | 0.0495 | 0.8419 | 0.1992 | 0.5198 |
| Cryptococcus | 0.1744 | 0.3681 | 0.1166 | 0.8499 | 0.1761 | 0.2822 | 0.2582 | 0.3239 |
| Sporobolomyces | 0.1928 | 0.3021 | 0.1550 | 0.6747 | 0.3922 | 0.0087 | 0.1855 | 0.5580 |
| Penicillium | 0.2658 | 0.1368 | −0.0431 | 0.9161 | 0.2382 | 0.1318 | 0.3978 | 0.0734 |
| Debaryomyces | 0.2311 | 0.2122 | 0.1786 | 0.6070 | 0.2472 | 0.1168 | 0.3613 | 0.1154 |
| Cystofilobasidium | 0.3799 | 0.0161 | 0.2369 | 0.4396 | 0.3607 | 0.0175 | 0.2835 | 0.2746 |
| Meyerozyma | 0.5835 | 3.207E-05 | 0.2167 | 0.5109 | 0.3348 | 0.0284 | 0.2247 | 0.4260 |

[a]n indicates the number of pairs of mothers and their children.
[b]Results of the main fungal genera (mean relative abundance > 1%) in samples of mothers and children are shown. Multiple Spearman's correlation tests of all fungal genera in children are performed to adjust the P value using the false discovery rate method.
[c]Rho values represent the strength of the Spearman's correlation.

younger group, the genus *Aspergillus* showed a significantly higher coefficient in the caesarean-born (rho = 0.4846, $P_{fdr}$ = 0.0127) than the vaginally born (rho = 0.2298, $P_{fdr}$ = 0.0144).

## DISCUSSION

In this study, we analyzed the differences in skin mycobiomes from the face, ventral forearm, and calf of 72 prepubescent children as well as the relationship between their skin mycobiome and that of their mothers to explore the influences of extrinsic and maternal factors on the children's cutaneous mycobiome. Common fungi detected globally in healthy human skin (34 to 37), including *Malassezia*, *Cladosporium*, *Alternaria*, *Candida*, and *Rhodotorula*, were present in the samples from all children in this study; this suggests that these fungal genera could represent important members of the skin mycobiome in prepubescent children. Similar to a previous report that analyzed the skin mycobiome of 13 children aged 7 to 14 years (38), in the current study, the genus *Malassezia* was less frequent in samples collected from the faces of children compared to their mothers. This may be attributed to the lack of sebaceous gland secretion in children's skin (39) and the obligatory lipophilic characteristic of *Malassezia* (40).

The differentiation of the skin mycobiome is positively correlated with age (41) and is less affected by sampling sites in children compared to their mothers. In children, the Shannon index was lower, whereas the individual specificity was higher for samples collected from the calf than those from the other sites; as human contact represents the primary route of fungal dispersal on the skin (42), these results indicate that skin sites with a high external environmental exposure, such as the face and ventral forearm, are more likely to be diverse and similar to the skin mycobiome of others. Unlike the results of our reported study on the cutaneous bacteria of children (32), the sampling sites in the current study minimally affected the skin microbiome diversity, with skin physiology being weakly correlated with the skin mycobiome. Additionally, the Sloan neutral model exhibited a much higher migration rate (m) in cutaneous bacteria than in fungi. Considering that the migration rate represents the probability that a random loss of an individual in a local community will be restored by dispersal from the metacommunity source, the results suggest a higher and more unhindered dispersal of cutaneous bacteria and a stronger dispersal limitation of cutaneous fungi.

In infants, the delivery mode largely determines the microbiomes of skin, nasal, oral, and fecal sites (18, 29 to 31, 43). However, whether the delivery mode has a long-term impact on the microbiome and host health of children remains controversial. In our previous study, the bacterial population, within the same cohort as described in the current study, differed in 10-year-old children according to different delivery modes (32). In comparison, in this study, the delivery mode was the most significant factor (excepting the children's age) affecting fungal composition, rather than sampling sites in cutaneous bacteria (32). The "Anna Karenina principle" suggests that dysbiotic individuals (people with an imbalanced human microbiome) have a more various microbial community composition than healthy individuals (44); consistent with this finding, caesarean-born children had a higher dispersion based on the principal coordinate analysis (PCoA) results, indicating the possibility of their deviation from a natural skin mycobiome. Based on the Sloan prediction model, we observed that the skin mycobiomes of caesarean-born children were assembled by more niche-based processes than those of vaginally born children. The previously drawn hypothetical association suggesting that an increase in influential niche-based processes in megacities is associated with skin diseases (45) may indicate that the observed increased risk of skin infection in caesarean-born children (24) is related to the closer-fitted niche-based process in caesarean-born children. Similarly, the maturation trajectories differed between the delivery modes, as the mycobiome of caesarean-born children exhibited no linear associations between the individual specificity and children's age in all sample groups. In contrast, vaginally born children showed positive linear associations in most groups, except for the calf site, which generally showed the same pattern, even after excluding the interference of sequence filtering and rarefaction (Fig. S7) as well as the interference from an uneven distribution of the delivery mode for samples from children aged 4 and 10 (Fig. S8). This lack of differentiation trajectories may be associated with increased influential niche-based processes in caesarean-born children. Furthermore, compared with those of the vaginally born children, the network density and transitivity of the skin mycobiome at all three sampling sites and richness (Chao1 index) in the face samples were low in caesarean-born children. Collectively, these results highlight the distinct dynamics and fragile community of the skin mycobiome of prepubescent children born via caesarean section.

The delivery mode also influences the fungal composition of children's skin. In 1-year-old children, the mean relative abundances of *Candida* and *Rhodoturula*, the main fungi inhabiting the vagina (46, 47), were higher in vaginally born children than in caesarean-born children; the skin-associated fungal genus *Malassezia* (35) and airborne genera *Alternaria* and *Aspergillus* (42, 48) exhibited the opposite trend. Based on the correlation analysis between children and mothers, the vagina-associated genera *Candida* and *Rhodotorula* were significantly correlated between vaginally born children from the younger group aged 1 to 3 and their mothers. *Candida* especially showed the strongest coefficient from the calf site (Table S5), whereas the same association was not observed for caesarean-born children. In caesarean-born mother–child pairs, the genus *Aspergillus* exhibited a significantly stronger correlation in the younger group, and the genera *Malassezia* and *Alternaria* exhibited this trend in the elder group. Ward et al. (31) reported that the genus *Candida* was dominant in the skin of vaginally born newborns; the number of *Candida*-reactive T cells in humans exceeds the number of those directed against other commensal or environmental fungi (4). Moreover, given the stability of *Candida* and the inconsistency of *Malassezia* and *Aspergillus* in infants aged 6 to 12 weeks (33), we postulate that vagina-associated commensal fungi transmitted from maternal sources have a more significant role in determining the skin mycobiome of vaginally born newborns and aiding the development of skin mycobiome dynamics via interacting with the host immune systems from infancy to childhood. In contrast, the skin mycobiomes of children born with exoteric interruption (caesarean section in the current study) were predominated by skin-associated and airborne fungi, resulting in a more static maturation trajectory compared to vaginally born children as they grew. Immune-imprinting events (the ability of microbes to tune immune composition or function) are active in early life, and some cannot be

reproduced later in life (17). These results suggest that the higher abundance of vagina-associated fungi in vaginally born children could be related to more influential neutral assembly processes and a more stable microbial network in the skin mycobiome compared with caesarean-born children.

In recent decades, caesarean deliveries have steadily increased, reaching 18.6% of births worldwide (49) and 34.9% in China (50). The increased incidence of several diseases in people born by caesarean delivery highlights the importance of characterizing the potential contributing factors. As ecological theory is required to fully understand the relationship between complex human microbiomes and human hosts (51), research regarding the long-term effect of delivery modes on the skin mycobiome of children may improve our understanding of the higher prevalence of diseases in caesarean-born children. Moving forward, it is necessary to conduct further investigations with larger and more comprehensive data sets, comprising subjects with different initial exposures from infancy to childhood, to more effectively define host–microbe interactions in early-life skin.

## MATERIALS AND METHODS

**Study design.** The study design, participant recruitment, and sample collection have been described previously (32). The Ethics Committee of Fudan University (Shanghai, China) approved the study procedures before implementation (Institutional Review Board number 2013-04-0446). This study was a single-visit cross-sectional study in children of different ages (1 to 5 and 10 years) performed from December 2012 to January 2013 in Shanghai, China. Participating mothers were required to complete an institutional review board-approved informed consent agreement in Mandarin and sign an agreement on their child's behalf before sample collection. All participant information was anonymized before analysis. The participants did not bathe for 12 h or apply hygiene products for 24 h before sampling. Children with a history of dermatologic disease as well as those who had received antibiotic treatment within 6 months preceding the study or were exposed to pets at home were excluded. Data on children's skin parameters, including moisture, transepidermal water loss (TEWL), and pH values of the face and ventral forearm have been reported previously (11, 32).

**Sample collection and preparation.** The exclusion criteria for subjects included a history of chronic dermatologic diseases, antibiotic treatment up to 6 months before sampling, and exposure to pets at home. The face (center of the cheek), ventral forearm (one quarter the length of the forearm from the hand), and calf (center of calf) regions (randomly assigned left or right) were sampled using sterile swabs as described previously (32). The swabs were first soaked in a solution of 0.15 M NaCl and 0.1% Tween 20, applied with moderate pressure, rolled back and forth 10 times over a $2 \times 2$-cm sampling region, removed with sterilized tweezers, and then placed in a PowerBead Tube of a PowerSoil DNA kit (Qiagen, Hilden, Germany). The total DNA was extracted according to the manufacturer's instructions and stored at $-20°C$ before PCR amplification. Owing to the low microbial biomass of the skin, we used nested PCR to amplify the fungal internal transcribed spacer 1 (ITS1) genes as described previously (33). NSA3 (5′-AAA CTC TGT CGT GCT GGG GAT A-3′) and NLC2 (5′-GAG CTG CAT TCC CAA ACA ACT C-3′) were used as primers for the first round of amplification (52, 53), and NSI1 (5′-NNN NNNNNN NNN GAT TGA ATG GCT TAG TGA GG-3′) and 58A2R (5′-NNN NNN NNN NNN CTG CGT TCT TCA TCG AT-3′) with 12-nt (nucleotide) barcodes were used as primers for the second round of amplification (52, 54). For the first round of nested PCR, each 25-$\mu$L reaction contained 12.5 $\mu$L Ex Taq Premix v.2.0 (TaKaRa, Dalian, China), 1 $\mu$L of each forward and reverse primer (2.5 $\mu$M), 5 $\mu$L template DNA, 1 $\mu$L bovine serum albumin (20 mg/mL), and 4.5 $\mu$L ddH$_2$O. PCR conditions were as follows: 94°C for 5 min, then 30 cycles at 94°C for 30 s, 50°C for 45 s, and 72°C for 60 s, with a final step at 72°C for 10 min. The second round was similar to the first round, except for 2.5 $\mu$L of product from the first round being used as the template, and the number of amplification cycles was 25. Negative controls for the PCR were conducted simultaneously with PCR amplification of all DNA samples. PCR products were pooled and purified using the UltraClean PCR cleanup kit (Qiagen). Libraries for sequencing of purified PCR products were constructed using the KAPA LTP Library kit (KAPA Biosystems, Boston, MA), according to the manufacturer's instructions, and sequenced on a Hiseq platform (Illumina, San Diego, CA) to generate $2 \times 250$-bp paired-end reads.

**Sequence analysis and microbial community characterization.** The quality of the sequencing data was evaluated using FastQC (https://www.bioinformatics.babraham.ac.uk/projects/fastqc/), and further sequence analysis and mycobiome community characterization were performed using Quantitative Insights into Microbial Ecology (QIIME) version 1.9 (55). First, the orientation of pair-end reads was adjusted according to the primer and barcode sequence. To prevent effects caused by variable lengths of PCR products from different fungi, oriented reads were trimmed to remove 12-nt barcodes (54) and retain only the first 200 nt with a Phred score of <20 in the last 50 nt. Sequences were assembled by placing the reverse complement of Read 2 in front of Read 1, assigned to samples according to the barcode, which was then gathered into one data set and chimera-checked against the UNITE database using USEARCH61 (http://drive5.com/usearch/usearch_docs.html). Next, sequences were clustered into OTUs according to a 95% similarity cutoff with USEARCH61 using the QIIME script "pick_open_reference_otus.py." OTUs with a fraction of total observations of <0.01% were discarded as spurious sequences (56), along with global singleton OTUs (appearing only in a single sample). Alpha diversity was assessed using

the Shannon and Chao1 indices, which evaluate the evenness and richness of the community, respectively. Beta diversity was assessed using the Bray-Curtis dissimilarity, which simultaneously evaluates community membership (presence of taxa) and composition (relative abundance of taxa). Both alpha and beta diversities were assessed at a depth of 2,500 sequences per sample. To confirm the principal contributor, samples were rarefied after being collapsed by children's age, delivery mode, and sampling site, and used for PCoA.

Skin bacterial communities from the same batch of participants were obtained from a previous study (32). Bacterial OTUs from previous studies were screened to identify 198 qualified children samples intersecting skin bacterial and fungal communities. Bacterial OTUs with a fraction of total observations of <0.01% were discarded, along with singleton OTUs.

**Statistical analysis.** Permutational multivariate analysis of variance (PERMANOVA) was applied to analyze the strength and significance of sample groupings by inputting the Bray-Curtis dissimilarity matrix using "compare_categories.py" in QIIME. The relationship between skin physical parameters and fungal abundance composition was evaluated with the Mantel test using "compare_distance_matrices.py." Pearson's correlation analysis was performed to identify associations between cutaneous physicochemical parameters and the relative abundance of the main fungal genera (mean relative abundance > 1%), as well as the age association of relative abundance of all fungal genera, bacterial, and fungal beta diversity indices in children. Spearman's rank correlation analysis was used to determine the associations between fungal genus taxa in children and their mothers. The Wilcoxon rank-sum test was used to assess differences in the mean fungal relative abundances, alpha and beta diversity indices. To avoid false positives, $P$ values for Wilcoxon rank-sum test (differences in fungal abundances), Spearman's correlation (correlation between fungal genera in mothers and children), Pearson's correlation (including age association and skin physical parameter association of fungal genera), and the Mantel test (correlation between skin physical parameters and fungal abundance composition) were corrected using the false discovery rate (FDR) method (57); adjusted $P$ values were obtained using the "p.adjust" function in the "stats" R package (https://www.rdocumentation.org/packages/stats/versions/3.6.2). Pearson's correlation analysis, Spearman's correlation analysis, and Wilcoxon rank-sum test were all performed using the "stats" R package. All visualizations were generated in R as well.

**Sloan neutral model prediction.** To determine the potential importance of stochastic processes in community assembly, the Sloan neutral model, developed for microbial communities based on metasequencing data (58), was used to predict the relationship between the OTU frequency and their relative abundance. The Sloan model evaluates whether the community assembly process from a metacommunity follows a stochastic model (within the 95% confidence interval of model predictions, as neutral) or a niche-based deterministic process (outside the 95% confidence interval of model predictions, as above or below) by testing all individual species in the metacommunity. The statistic $R^2$ represents the overall fit to the neutral model, and the estimated migration rate (m) represents a proxy for dispersal limitation. Sloan neutral model prediction and statistics were performed using the "MicEco" package in R.

**Ecological network construction.** In the network analysis of the skin mycobiome, fungal OTUs with a frequency of <30% were excluded from the samples at each sampling site. A total of 130, 122, and 100 OTUs were used in the network analyses of the face, ventral forearm, and calf, respectively. The SPIEC-EASI framework (59), a statistical method that enables the inference of microbial ecological networks from OTU data sets, was performed as described previously (45). We ran the network analysis using the neighborhood algorithm (60) and the StARS (Stability Approach to Regularization Selection) method (61) with a minimum $\lambda$ threshold of 0.1% and a total of 30 penalties. All network construction steps were computed using the R package "SpiecEasi" (version 1.0.7).

**Data availability.** Raw sequence data are available in the National Omics Data Encyclopedia (NODE, https://www.biosino.org/node/index) under the project ID OEP003185. The Strengthening The Organization and Reporting of Microbiome Studies (STORMS) checklist is available at https://doi.org/10.5281/zenodo.6624990.

## SUPPLEMENTAL MATERIAL

Supplemental material is available online only.
**SUPPLEMENTAL FILE 1**, PDF file, 2.3 MB.

## ACKNOWLEDGMENTS

This study was funded by Johnson & Johnson International Pte. Ltd. (Singapore). Zhe-Xue Quan was supported by the Shanghai Municipal Science and Technology Major Project (Grant No. 2017SHZDZX01) and National Key R&D Program of China (2021YFA1301000). We thank all the study participants.

Yuan-Yuan Duan, Fan-Qi Kong, and Carlos Galzote were employed by Johnson & Johnson. The remaining authors declare that the research was conducted in the absence of any commercial or financial relationships that could be construed as potential conflicts of interest. We declare that this study received funding from Johnson & Johnson International Pte Ltd. (Singapore). The funder was not involved in the study design, collection, analysis, interpretation of data, writing of this article, or the decision to submit it for publication.

Conceptualization: Z.-X.Q., F.-Q.K., Y.-Y.D., C.G.; Data Curation: Y.-R.W., T.Z.; Formal Analysis: Y.-R.W., T.Z., Z.-X.Q., F.-Q.K.; Funding Acquisition: Z.-X.Q., C.G.; Investigation: Y.-R.W., T.Z., F.-Q.K., Y.-Y.D.; Methodology: Z.-X.Q., C.G.; Project Administration: Z.-X.Q., C.G.; Resources: F.-Q.K., Y.-Y.D., C.G.; Software: Y.-R.W., T.Z.; Supervision: Z.-X.Q.; Validation: Y.-R.W., T.Z., Z.-X.Q., C.G.; Visualization: Y.-R.W., T.Z.; Writing – Original Draft: Y.-R.W., T.Z., Z.-X.Q., C.G.; Writing – Review and Editing: Y.-R.W., T.Z., Z.-X.Q., C.G.

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
