## [Reviewer comments · Microbiology Spectrum]

Microbiology Spectrum

Infant mode of delivery shapes the skin mycobiome of prepubescent children

Yan-Ren Wang, Ting Zhu, Fan-Qi Kong, Yuan-Yuan Duan, Carlos Galzote, and Zhe-Xue Quan

Corresponding Author(s): Zhe-Xue Quan, Fudan University

Review Timeline:

Submission Date:	June 17, 2022
Editorial Decision:	July 9, 2022
Revision Received:	August 4, 2022
Accepted:	August 17, 2022

Editor: Jan Claesen

Reviewer(s): Disclosure of reviewer identity is with reference to reviewer comments included in decision letter(s). The following individuals involved in review of your submission have agreed to reveal their identity: Laura Tipton (Reviewer #1)

Transaction Report:

DOI: <https://doi.org/10.1128/spectrum.02267-22>

July 9, 2022

Prof. Zhe-Xue Quan
Fudan University
2005 Songhu Road
Shanghai
China

Re: Spectrum02267-22 (Infant mode of delivery shapes the skin mycobiome of prepubescent children)

Dear Prof. Zhe-Xue Quan:

Thank you for submitting your research to Spectrum. Your manuscript has been evaluated by two independent Reviewers, who are both enthusiastic about your study (as am I)! The Reviewers raised some points that would help improve your manuscript. I would be happy to consider a revised version of your paper, addressing all Reviewer comments in a point-by-point fashion.

Link Not Available

Sincerely,

Jan Claesen

Journals Department
Reviewer comments:

Reviewer #1 (Comments for the Author):

In this manuscript, the authors present a broad analysis of the skin mycobiome in children aged 1-10 years old. Using ITS target amplicon sequences, they present composition, diversity, assembly models, network analyses, specificity, differences between birth modes, and dissimilarity from the child's mother. The authors find that delivery mode has the strongest association with multiple microbiome properties. Overall it is well written and covers all the analyses I like to see in a new micro- or mycobiome study.

My main concerns are as follows:

1. Was anything done to account for samples that originate from the same individual? Particularly in the PERMANOVA tests based on demographics such as age, delivery mode, birthplace where all sampling sites were included. If accounted for, it should be clearly stated in the results or methods sections.
2. It is unclear how many samples were used in each Sloan model and SPIEC-EASI network shown in figure 2. If I have done the math correctly, based on the numbers in table 1, there are 44 samples in each vaginally-born model and network, and 28 in each caesarean-born model and network. These are low numbers for network analyses which tend to do best with over 50 samples. Options to compensate for these low numbers may include increasing the StARS lambda threshold to 0.1%, or building a combined network and pulling out neighborhoods of taxa that are enriched in each population of interest (ie the face of caesarean-born children). Otherwise, I would request that the low sample size be mentioned in the discussion section.
3. "No significance" is used throughout the manuscript based on a p-value <0.05, however, this value is arbitrary and I want to see F-values or p-values, regardless of if they are "significant" or not.

Other minor issues include:

4. Birthplace is mentioned on line 96 but not defined
5. The Sloan neutral model is well explained in the section of delivery mode (lines 133-135), however, it was first mentioned in the section on sampling sites (lines 114-124). It would be helpful to move the explanation up to the first time the model is mentioned.
6. Within the discussion, I found lines 206-207, discussing *Malassezia* enrichment in girls to be unnecessary and distracting since sex differences were not discussed in the results section.
7. On line 220, should be "dispersal from the metacommunity source" rather than sink.
8. On lines 231-232 "caesarean-born children had a higher dispersal based on the PCoA results, indicating deviation from a healthy skin mycobiome." makes it sound like c-section children are inherently dysbiotic but I don't see anything beyond their higher dispersion to support this.
9. Are the results the same if you do not rarefy to 2,500 sequences per sample? This is of course a widely debated practice but see McMurdie and Holmes, 2014 and Hong, et al 2022 for discussion.
10. There are several places, particularly in the tables and figures that refer to Bray-Curtis as a distance but it is a dissimilarity, as correctly stated in figure 3.

Reviewer #2 (Comments for the Author):

The authors attempted to elucidate the mycobiome of prepubescent children by internal transcribed spacer amplicon sequencing. Furthermore, the authors revealed the diversity of each group and relationships between children's and mother's mycobiome by using various statistical analyses. Sloan neutral model prediction and ecological network construction were also conducted to compare the mycobiome community of vaginally born and caesarean-born children. This manuscript is interesting, but I would like to make a few comments.

Major comments

1. Sloan neutral model prediction used in this manuscript was originally used for prokaryotes. It is not known whether this predictive model also applies to eukaryotes. Furthermore, Kim et al., 2018 cited by the author at lines 234-236 is also research about prokaryotes. Any studies or evidence are needed that Sloan neutral model can also be applied to research for eukaryotes.
2. The most of Rho values in Table 2 is too low to claim correlations between mycobiome of children and mother. Even if the p value is low enough, it is difficult to claim that Rho values lower than 0.5 are meaningful.
3. The authors claimed that clear intra- and interspecialization of fungal structure was not observed in caesarean-born children by using Bray-Curtis dissimilarity. However, numbers of samples of caesarean-born children seem not enough. For example, only 3 and 2 samples are existed in 4 and 10 year caesarean-born children group, respectively.
4. In Figure 5, the abundance of the genus *Malassezia* was much higher in mothers that older than their children, and authors also discussed about this at line 201-206. However, mean relative abundance of the genus *Malassezia* is negatively correlated with ages in both delivery groups in Figure 4b. why?

Minor comments

1. Line 114-124: This paragraph seems not appropriate in the title. I recommended to move this paragraph to another place or make new title.
2. Line 154-155: R and p values don't match to figure 3A.
3. Line 206-207: Please explain the reason why *Malassezia* was significantly enriched in girls born in suburban.

Staff Comments:

Preparing Revision Guidelines

Please return the manuscript within 60 days; if you cannot complete the modification within this time period, please contact me. If you do not wish to modify the manuscript and prefer to submit it to another journal, please notify me of your decision immediately so that the manuscript may be formally withdrawn from consideration by Microbiology Spectrum.

Dear Editor,

Based on the reviewers' comments and suggestions, we have carefully revised the manuscript and have provided point-by-point responses. We hope that our revised manuscript will be suitable for publication in *Microbiology Spectrum*.

Zhe-Xue Quan

Reviewer Comments:

Reviewer #1

In this manuscript, the authors present a broad analysis of the skin mycobiome in children aged 1-10 years old. Using ITS target amplicon sequences, they present composition, diversity, assembly models, network analyses, specificity, differences between birth modes, and dissimilarity from the child's mother. The authors find that delivery mode has the strongest association with multiple microbiome properties. Overall it is well written and covers all the analyses I like to see in a new micro- or mycobiome study.

RESPONSE: We thank the reviewer for these encouraging comments.

Major comments:

1. Was anything done to account for samples that originate from the same individual? Particularly in the PERMANOVA tests based on demographics such as age, delivery mode, birthplace where all sampling sites were included. If accounted for, it should be clearly stated in the results or methods sections.

RESPONSE:

In the result of PERMANOVA (Table 1), the compound row of the "Sampling site" accounted for three types of samples from each individuals (n = numbers of participant children = 73 individuals).

Furthermore, the children's age, delivery mode, and birthplace were more influential than sampling sites on the skin mycobiome in this study; hence, we mainly discussed the PERMANOVA tests based on samples where all three sampling sites were included (Lines 94-97, Page 6, Lines 129-131, Page 7).

2. It is unclear how many samples were used in each Sloan model and SPIEC-EASI network shown in figure 2. If I have done the math correctly, based on the numbers in table 1, there are 44 samples in each vaginally-born model and network, and 28 in each caesarean-born model and network. These are low numbers for network analyses which tend to do best with over 50 samples. Options to compensate for these low numbers may include increasing the StARS lambda threshold to 0.1%, or building a combined network and pulling out neighborhoods of taxa that are enriched in each population of interest (ie the face of caesarean-born children). Otherwise, I would request that the low sample size be mentioned in the discussion section.

RESPONSE: We thank the reviewer for this suggestion and have performed the

normatively modified SPIEC-EASI network as described (lambda threshold to 0.1%). We have also changed the nlambda from 20 to 30, as the former parameter could not sample the lambda path finely. The modified network does not differ from the former one substantially (in that the samples from vaginally-born children form a more stable network than that of the caesarean-born).

The corresponding change has been made in Figure 2b and the text of results (Lines 144-147, Page 8): "face (vaginally-born: D = 0.013, T = 0.333, caesarean-born: D = 0.008, T = 0.148), ventral forearm (vaginally-born: D = 0.015, T = 0.538, caesarean-born: D = 0.014, T = 0.297), and calf (vaginally-born: D = 0.025, T = 0.242, caesarean-born: D = 0.022, T = 0.192).", and also the text of methods (Lines 406-407, Page 20): "We ran the network analysis using the neighborhood algorithm (60) and the StARS (Stability Approach to Regularization Selection) method (61) with a minimum λ threshold of 0.1% and a total of 30 penalties. All network construction steps were computed using the R package "SpiecEasi" (version 1.0.7).".

3. "No significance" is used throughout the manuscript based on a p-value <0.05, however, this value is arbitrary and I want to see F-values or p-values, regardless of if they are "significant" or not.

RESPONSE: Thank you for your suggestion. We have already exhibited the F-values for the PERMANOVA result (Table 1) and P_{fdr} -values for the difference test of relative fungal abundances (Figure S5) and the Spearman's correlation (Table 2, S3 and S4).

Note that regarding the examination of the differences in relative abundance, we chose the "Stat" package and "p.adjust" function in R rather than "group_significance.py" in QIIME1; we also changed the examination method from "t.test" to the more conservative "wilcox.test" (in Figure 4b and S5). Similarly, regarding the Spearman's correlation test of fungal genera between mothers and their children (Figure S6; Table 2 and S5) and the Pearson's correlation test of age association with fungal genera (Table S4), to mitigate the increasing chance of a type I error depending on the raw P -value threshold, we subjected all fungal genera (instead of only major genera) to perform multiple Spearman's correlation tests (between mothers and their own children) and multiple Wilcoxon tests; and further applied a p-value adjustment with the "FDR" method. The corresponding change has been made in text of methods (Lines 377-385, Pages 18-19): "The Wilcoxon rank-sum test was used to assess differences in the mean fungal relative abundances, alpha and beta diversity indices. To avoid false positives, P -values for Wilcoxon's rank-sum test (differences in fungal abundances), Spearman's correlation (correlation between fungal genera in mothers and children), Pearson's correlation (including age association and skin physical parameter association of fungal genera), and the Mantel test (correlation between skin physical parameters and fungal abundance composition) were corrected using the false discovery rate (FDR) method (57); adjusted P -values were obtained using the "p.adjust" function in the "stats" R

package (<https://www.rdocumentation.org/packages/stats/versions/3.6.2>).".

The edited sentences are as follows:

In Lines 129-131 (Page 7), details on the F-values of PERMANOVA with P -values > 0.05 were added: "The skin mycobiomes of caesarean-born children did not differ significantly between sampling sites (vaginally-born: pseudo-F = 1.482, $P \leq 0.05$, caesarean-born: pseudo-F = 0.906, $P > 0.05$) or feeding type (vaginally-born: pseudo-F = 1.447, $P \leq 0.05$, caesarean-born: pseudo-F = 1.140, $P > 0.05$), whereas significant differences were observed in vaginally-born children (Table 1)."

In Lines 168-174 (Page 9), information on reanalyzed P_{fdr} -values of the Wilcoxon's test on differences of fungal genera was added: "Compared with caesarean-born children, one-year-old vaginally-delivered children exhibited greater relative abundances of several main fungal genera (Fig. S5): *Rhodotorula* (vaginally-born: 15.73%, caesarean-born: 5.11%, $P_{fdr} = 0.2829$), *Candida* (vaginally-born: 10.97%, caesarean-born: 3.55%, $P_{fdr} = 0.1700$), and *Cladosporium* (vaginally-born: 10.52%, caesarean-born: 2.77%, $P_{fdr} = 0.0392$). However, certain genera showed the opposite trend, namely *Malassezia* (vaginally-born: 13.31%, caesarean-born: 22.61%, $P_{fdr} = 0.4319$), *Alternaria* (vaginally-born: 4.78%, caesarean-born: 10.43%, $P_{fdr} = 0.4631$), and *Aspergillus* (vaginally-born: 2.86%, caesarean-born: 6.07%, $P_{fdr} = 0.8824$).". This value has also been added to Figure S5.

In Lines 175-177 (Page 9), the reanalyzed R- and P_{fdr} -values of Pearson's correlation test between fungal genera and children's age were added: "Furthermore, as the children's ages increased, *Candida* (vaginally-born: R = -0.2248, $P_{fdr} = 0.1432$, caesarean-born: R = -0.0971, $P_{fdr} = 0.9206$) showed a more substantial trend of decrement in vaginally-born children than in caesarean-born children (Table S4)". This changes has also been made in Table S4.

In Line 189-190 (Page 10), the reanalyzed P_{fdr} -value of Wilcoxon's test of the differences of *Malassezia* has been added: "Significant differences were observed in the relative skin mycobiome compositions of mothers and children for genus *Malassezia* at the face site (mothers: 27.85%, children: 15.56%, $P_{fdr} = 0.0011$, Fig. 4b)". This value has also been added to Figure 4b.

Note, as we re-analyzed and re-examined the statistical tests (Wilcoxon's test: Figure 4b and S5, Pearson's correlation test: Table S4, Spearman's correlation test: Figure S6; Table 2 and S5) mentioned above for all fungal genera in children or children and mothers (instead of only the major genera as done previously) to perform the p-value adjustment, slight changes occurred in the relative abundance/R-value/rho-value with no substantial impact; these corresponding changes have been made in the text of results; figures and tables mentioned above and also their captions.

Minor comments:

1. Birthplace is mentioned on line 96 but not defined.
RESPONSE: The definition of birthplace (and delivery mode) has been added to the text (Lines 96-97, Page 6): "followed by delivery mode (pseudo-F = 2.136, $P \leq 0.05$, including children delivered through vaginal birth and caesarean section), and birthplace (pseudo-F = 2.051, $P \leq 0.05$, including children born in suburban and urban areas)."
2. The Sloan neutral model is well explained in the section of delivery mode (lines 133-135), however, it was first mentioned in the section on sampling sites (lines 114-124). It would be helpful to move the explanation up to the first time the model is mentioned.
RESPONSE: Thank you for your suggestion. We have moved the explanatory sentences to Lines 118-122 (Page 7): "The bacterial communities on skin were more influenced by dispersal (higher m values, fungi: $m = 0.048$, bacteria: $m = 0.491$, Fig. S2a) and showed a higher proportion of OTUs above the neutral prediction compared to fungi, although they were less fitted to the neutral model (fungi: $R^2 = 0.501$, bacteria: $R^2 = 0.489$, Fig. S2a)."
3. Within the discussion, I found lines 206-207, discussing *Malassezia* enrichment in girls to be unnecessary and distracting since sex differences were not discussed in the results section.
RESPONSE: Thank you for your suggestion. We have deleted this distracting result (texts and the previous Figure S5) from the discussion.
4. On line 220, should be "dispersal from the metacommunity source" rather than sink.
RESPONSE: Thank you for your suggestion. We have changed this description in the text (Line 229-230, Page 12): "a random loss of an individual in a local community will be restored by dispersal from the metacommunity source"
5. On lines 231-232 "caesarean-born children had a higher dispersal based on the PCoA results, indicating deviation from a healthy skin mycobiome." makes it sound like c-section children are inherently dysbiotic but I don't see anything beyond their higher dispersion to support this.
RESPONSE: Considering that this misunderstanding could be caused by our choice of words and degree of tone, we have modified this part of the discussion in Line 241-242 (Page 12): "consistent with this finding, caesarean-born children had a higher dispersion based on the PCoA results, indicating the possibility of their deviation from a natural skin mycobiome."
6. Are the results the same if you do not rarefy to 2,500 sequences per sample? This is of course a widely debated practice but see McMurdie and Holmes, 2014 and Hong, et al 2022 for discussion.

RESPONSE: Thank you for your suggestion. As mentioned by McMurdie and Holmes, the rarefaction of the library size functions as moderating the additional observation of rare OTUs brought by the effects of library size differences. A study recently claimed that the rare taxa in amplicon sequencing could be spurious sequences that may markedly influence the outcome of later microbiome analyses (Reitmeier et al., *ISME Communications* 2021, <https://doi.org/10.1038/s43705-021-00033-z>). Therefore, in this study, prior to the rarefaction to 2,500 sequences per sample, we applied the filtering strategy of the rare taxa (minimum relative abundance threshold at 0.01%, mentioned in Lines 354-355). Additionally, Hong *et al.* indicated that although the rarefaction could generate different degrees of sensitivity loss, it still could be a valuable tool in guaranteeing the validity of the later permutation tests.

We also added the analysis of dissimilarity (across ages between children from different delivery types) without filtering and rarefying, shown in Figure S7. This result is identical to the results obtained after rarefying (Figure 3). The relevant changes have been added to Lines 252-254 (Page 13): "By contrast, vaginally-born children showed positive linear associations in most groups, except for the calf site, which generally showed the same pattern, even after excluding the interference of sequence filtering and rarefaction (Fig. S7) as well as the interference from an uneven distribution of the delivery mode for samples from children aged 4 and 10 (Fig. S8).".

7. There are several places, particularly in the tables and figures that refer to Bray-Curtis as a distance but it is a dissimilarity, as correctly stated in figure 3.

RESPONSE: Thanks for your suggestion. We have changed this description (from Bray-Curtis distance to Bray-Curtis dissimilarity) in the captions of Table 1 and Figure 4.

Reviewer #2

The authors attempted to elucidate the mycobiome of prepubescent children by internal transcribed spacer amplicon sequencing. Furthermore, the authors revealed the diversity of each group and relationships between children's and mother's mycobiome by using various statistical analyses. Sloan neutral model prediction and ecological network construction were also conducted to compare the mycobiome community of vaginally born and caesarean-born children. This manuscript is interesting, but I would like to make a few comments.

RESPONSE: We thank the reviewer for the encouraging comments.

Major comments:

1. Sloan neutral model prediction used in this manuscript was originally used for prokaryotes. It is not known whether this predictive model also applies to eukaryotes. Furthermore, Kim et al., 2018 cited by the author at lines 234-236 is also research about prokaryotes. Any studies or evidence are needed that Sloan neutral model can also be applied to research for eukaryotes.

RESPONSE: Some studies have already used the Sloan neutral model prediction for eukaryotes, for example, in the field of human skin mycobiomes; these include Leung et al., *Microbiome* 2020 (<https://doi.org/10.1186/s40168-020-00874-1>) and Tong et al., *mSystems* 2019 (<https://doi.org/10.1128/mSystems.00004-19>). This model has also been used in a study of mycobiomes of other animals (stonefile): Zhu et al., *npj biofilms and microbiomes* 2022 (<https://doi.org/10.1038/s41522-022-00298-9>), as well as airborne mycobiomes (from human exhaled breath condensate): Zhang et al., *Environmental Science and Technology* 2022 (<https://doi.org/10.1021/acs.est.2c00688>).

2. The most of Rho values in Table 2 is too low to claim correlations between mycobiome of children and mother. Even if the p value is low enough, it is difficult to claim that Rho values lower than 0.5 are meaningful.

RESPONSE: This problem may occur because of three reasons: 1) Skin fungi could be more resistant to migration owing to its low migration rate compared to skin bacteria; thus, in this study, rho values of fungal genera are generally lower than those presented in our previous bacterial study (Zhu et al., 2019). 2) The total cohort includes children from various age groups; hence, with the maturation of children, these correlations of the initial influence might be hindered. 3) Using a combination of different sampling sites in the analysis expands the numbers of samples and weakens the coefficient of this maternal correlation.

According to the second assumption mentioned above, we sub-sampled the total cohorts based on the children's ages (i.e., a younger group of 1- to 3-year-olds comprising 35 individual children and their mothers, and an elder group of 4-, 5-, and 10-year-olds comprising 37 individual children and their mothers). Then, as described in the third major comment of reviewer #1, to mitigate increasing the chance of a type I error depending on the raw *P*-value threshold, we subjected all fungal genera into multiple Spearman's correlation tests between mothers and their own children and further applied a p-value adjustment using the "FDR" method. The result is shown in Table 2. As expected, maternal correlations of the vagina-associated genera (*Candida* and *Rhodotorula*) are significantly stronger in the vaginally-born children of the younger group than the elder. By contrast, except for the genus *Aspergillus*, the skin-associated and airborne genera (*Malassezia* and *Alternaria*) showed a stronger correlation in caesarean-born children of the elder group (*Aspergillus* was stronger in caesarean-born children of the younger group). The description of these results has thus been modified in Lines 190-205 (Pages 10-11): "Spearman's correlation test was then employed to reveal associations between the fungal genera abundance in mother-child pairs from two groups divided by children's age (Table 2 and Fig. S6, main genera with a mean relative abundance > 1% in all samples are presented). In groups with 1- to 3-year-old children, *Candida* ($\rho = 0.4067$, $P_{fdr} = 0.0088$) and *Rhodotorula* ($\rho = 0.3771$, $P_{fdr} = 0.0167$) showed a significant correlation between vaginally-born children and their mothers but not between caesarean-born children and their mothers ($\rho \leq 0.0505$, $P_{fdr} \geq 0.9161$). Furthermore, vaginal birth mother-child pairs from 4-, 5-, and 10-year-old children did not show significant correlations with respect to these two genera ($\rho \leq 0.1253$, $P_{fdr} \geq 0.4684$). Caesarean birth mother-child pairs from the elder age group exhibited positive relationships with respect to *Malassezia* ($\rho = 0.5158$, $P_{fdr} = 0.0104$) and *Alternaria* ($\rho = 0.4555$, $P_{fdr} = 0.0332$); likewise, these trends were not present in the vaginal birth pairs from the elder group ($\rho \leq 0.1306$, $P_{fdr} \geq 0.4640$) and the caesarean birth pairs from the younger group ($\rho \leq 0.2172$, $P_{fdr} \geq 0.5109$). Additionally, in mother-child pairs of the younger group, the genus *Aspergillus* showed a significantly higher coefficient in the caesarean-born ($\rho = 0.4846$, $P_{fdr} = 0.0127$) than the vaginally-born ($\rho = 0.2298$, $P_{fdr} = 0.0144$).", and also in Table 2; the

visualization of major result in Table 2 has been added up as Figure S6.

Furthermore, regarding the third assumption mentioned above, we analyzed the Spearman's correlations according to different sampling sites, age group and delivery mode; this information is shown in Table S5. Taking *Candida* in vaginally-born children from the younger group and *Alternaria* in caesarean-born children from the elder group as examples, rho-values of maternal correlation differ by sampling site. 1) *Candida*: the calf exhibited the strongest correlation ($\rho = 0.5520$, $P_{fdr} = 0.1205$), followed by face ($\rho = 0.3504$, $P_{fdr} = 0.4197$) and ventral forearm ($\rho = 0.2408$, $P_{fdr} = 0.6183$); 2) *Alternaria*: the ventral forearm exhibited the strongest correlation ($\rho = 0.7088$, $P_{fdr} = 0.0823$), followed by the calf ($\rho = 0.4780$, $P_{fdr} = 0.3487$) and face ($\rho = 0.3411$, $P_{fdr} = 0.6490$). These results indicated the differing degrees of impact of this maternal correlation at different skin sites.

The relevant description has been modified as follows:

Lines 25-29, Page 2: "The maternal correlation with children also differs based on the mode of delivery; specifically, the mycobiomes of vaginally-born children at younger age are more strongly correlated with vagina-associated fungal genera (*Candida* and *Rhodotorula*), whereas those of caesarean-delivered children at elder age include more skin-associated and airborne fungal genera (*Malassezia* and *Alternaria*).".

Lines 265-271, Pages 13-14: "Based on the correlation analysis between children and mothers, the vagina-associated genera *Candida* and *Rhodotorula* were significantly correlated between vaginally-born children from the younger group aged 1-3 with their mothers. *Candida* especially showed the strongest coefficient from the calf site (Table S5), whereas the same association was not observed for caesarean-born children. In caesarean-born mother-child pairs, the genus *Aspergillus* exhibited a significantly stronger correlation in the younger group, and the genera *Malassezia* and *Alternaria* exhibited this trend in the elder group.".

3. The authors claimed that clear intra- and inter specialization of fungal structure was not observed in caesarean-born children by using Bray-Curtis dissimilarity. However, numbers of samples of caesarean-born children seem not enough. For example, only 3 and 2 samples are existed in 4 and 10 year caesarean-born children group, respectively.

RESPONSE: The lack of samples from 4- and 10-year-old children delivered by caesarean was caused by the filtering of unqualified samples. However, this did not have conflict with the current results. We tested the dissimilarity analysis after deleting samples from the 4- and 10-year-old children (this result is shown in Figure S8). Generally, the dissimilarity of caesarean-born children all showed no significant linear correlation with age, and the increasing trends of dissimilarity in vaginally-born children were enriched except from face sites. This examination proves the validity of this part of the result.

A relevant change has been added to Lines 252-254 (Page 13): "By contrast, vaginally-born children showed positive linear associations in most groups, except

for the calf site, which generally showed the same pattern, even after excluding the interference of sequence filtering and rarefaction (Fig. S7) as well as the interference from an uneven distribution of the delivery mode for samples from children aged 4 and 10 (Fig. S8)."

4. In Figure 5, the abundance of the genus *Malassezia* was much higher in mothers that older than their children, and authors also discussed about this at line 201-206. However, mean relative abundance of the genus *Malassezia* is negatively correlated with ages in both delivery groups in Figure 4b. why?

RESPONSE: Thanks for your suggestion. The abundance of the genus *Malassezia* on human skin does not show the overall linear correlation throughout all ages: The contents of *Malassezia* declined in the elderly compared with the young (Wu et al., *mSphere* 2020). Another study reported that the abrupt increment of sebum occurred until age 13 (Pochi et al., *Journal of Investigative Dermatology* 1979) and the children in this study are all younger than 10 years old, indicating that the immature sebaceous glands in children herein cannot be compared to adult or pubertal children. This could be the reason why the relative abundance of *Malassezia* did not show a positive correlation with age in this study.

Note, as described in the third major comment of reviewer #1, to mitigate the false positive possibility, we also re-examined the Pearson's correlation results of all fungal genera (instead of only the major genera as done previously, thereby not substantially impacting the R-value) and further adjusted the *P*-value of Pearson's correlation between the fungal genera and children's ages. Only the genus *Fusarium* showed a positive correlation with age significantly in vaginally-born children. Therefore, we moved this part of the results from previous Figure 4b-e to Table S4 (and thus moved previous Figure 4a to Figure S5). The edited version of this text in the manuscript is shown as follows (Lines 175-177, Page 9): "Furthermore, as the children's ages increased, *Candida* (vaginally-born: $R = -0.2248$, $P_{fdr} = 0.1432$, caesarean-born: $R = -0.0971$, $P_{fdr} = 0.9206$) showed a more substantial trend of decrement in vaginally-born children than in caesarean-born children (Table S4).".

Minor comments:

1. Minor-1. Line 114-124: This paragraph seems not appropriate in the title. I recommended to move this paragraph to another place or make new title.

RESPONSE: Thank you for your suggestion. This paragraph has been separated into its own subsection (Line 115, Page 7): "Neutral assembly process on skin mycobiomes and bacteriomes".

2. Line 154-155: R and p values don't match to figure 3A.

RESPONSE: Thank you for your notification. These mismatches have been corrected in the revised manuscript (Line 158, Page 9): "site specificity (Fig. 3a) in vaginally-born children was more strongly correlated with age (vaginally-born: $R = 0.156$, $P = 0.085$, caesarean-born: $R = 0.102$, $P = 0.357$).".

3. Line 206-207: Please explain the reason why *Malassezia* was significantly enriched in girls born in suburban.

RESPONSE: Thank you for your notification. The detailed cause of this result is unclear to us. According to the suggestion of reviewer #1 (the third minor comment), we have already deleted this information (previously Figure S5) in the manuscript.

August 17, 2022

Prof. Zhe-Xue Quan
Fudan University
2005 Songhu Road
Shanghai
China

Re: Spectrum02267-22R1 (Infant mode of delivery shapes the skin mycobiome of prepubescent children)

Dear Prof. Zhe-Xue Quan:

Thanks for addressing the Reviewer comments. I would like to congratulate you on the acceptance of your study for publication in Spectrum!

Your manuscript has been accepted, and I am forwarding it to the ASM Journals Department for publication. You will be notified when your proofs are ready to be viewed.

Sincerely,

Jan Claesen
Editor, Microbiology Spectrum
